# Dichloroacetate-induced metabolic reprogramming improves lifespan in a Drosophila model of surviving sepsis

Veli Bakalov[1,2‡], Laura Reyes-Uribe[1‡], Rahul Deshpande[3], Abigail L. Maloy[1], Steven D. Shapiro[4], Derek C. Angus[1], Chung-Chou H. Chang[1,4,5], Laurence Le Moyec[6,7], Stacy Gelhaus Wendell[3], Ata Murat Kaynar[1,8]*

1 Clinical Research, Investigation, and Systems Modeling of Acute Illness (CRISMA) Laboratory, Department of Critical Care Medicine, University of Pittsburgh School of Medicine, Pittsburgh, PA, United States of America, 2 Medicine Institute, Allegheny Health Network, Pittsburgh, PA, United States of America, 3 Department of Pharmacology and Chemical Biology, University of Pittsburgh, Pittsburgh, PA, United States of America, 4 Department of Medicine, University of Pittsburgh, Pittsburgh, PA, United States of America, 5 Department of Biostatistics, Graduate School of Public Health, University of Pittsburgh, Pittsburgh, PA, United States of America, 6 Université d'Evry Val d'Essonne—Université Paris-Saclay, Evry, France, 7 Muséum National d'Histoire Naturelle, Unité MCAM, UMR7245 CNRS, Paris, France, 8 Department of Anesthesiology and Perioperative Medicine, University of Pittsburgh, Pittsburgh, PA, United States of America

‡ These authors contributed equally to this work.
* kaynar@pitt.edu

**Data Availability Statement:** All relevant data are within the manuscript and its Supporting Information files.

**Funding:** Dr. Kaynar received research grant support from NIH (HL126711). The funders had no

## Abstract

Sepsis is the leading cause of death in hospitalized patients and beyond the hospital stay and these long-term sequelae are due in part to unresolved inflammation. Metabolic shift from oxidative phosphorylation to aerobic glycolysis links metabolism to inflammation and such a shift is commonly observed in sepsis under normoxic conditions. By shifting the metabolic state from aerobic glycolysis to oxidative phosphorylation, we hypothesized it would reverse unresolved inflammation and subsequently improve outcome. We propose a shift from aerobic glycolysis to oxidative phosphorylation as a sepsis therapy by targeting the pathways involved in the conversion of pyruvate into acetyl-CoA via pyruvate dehydrogenase (PDH). Chemical manipulation of PDH using dichloroacetic acid (DCA) will promote oxidative phosphorylation over glycolysis and decrease inflammation. We tested our hypothesis in a *Drosophila melanogaster* model of surviving sepsis infected with Staphylococcus aureus. Drosophila were divided into 3 groups: *unmanipulated*, *sham* and *sepsis survivors*, all treated with linezolid; each group was either treated or not with DCA for one week following sepsis. We followed lifespan, measured gene expression of *Toll*, *defensin*, *cecropin A*, and *drosomycin*, and levels of lactate, pyruvate, acetyl-CoA as well as TCA metabolites. In our model, metabolic effects of sepsis are modified by DCA with normalized lactate, TCA metabolites, and was associated with improved lifespan of sepsis survivors, yet had no lifespan effects on unmanipulated and sham flies. While *Drosomycin* and *cecropin A* expression increased in sepsis survivors, DCA treatment decreased both *and* selectively increased *defensin*.

role in study design, data collection and analysis, decision to publish, or preparation of the manuscript. Dr. Kaynar received salary support from NIH (HL126711).

**Competing interests:** No author claims competing interest.

## Introduction

Advances in diagnostic modalities, prevention of complications, and care bundles improve sepsis-associated short-term mortality; however, sepsis remains a leading cause of death in hospitalized patients and beyond [1]. In addition to high mortality rates, sepsis survivors experience long-term complications, such as accelerated cardiovascular and neuro-cognitive decline, new infections, cancer, and metabolic disturbances [2, 3]. Importantly, late deaths among sepsis survivors are not solely explained by health status before sepsis, implying some feature of the sepsis itself contributes to these late sequelae [4–7].

During sepsis, immune cells shift their metabolic balance towards aerobic glycolysis over oxidative phosphorylation producing excessive amounts of lactate, a marker for sepsis severity, even under normoxic conditions [8–11]. The metabolic changes occurring during sepsis in turn lead to further immune cell activation and unresolved inflammation [12, 13]. Clinical [14] and laboratory [13, 15–17] work suggests a link between unresolved inflammation and long-term sequelae making the aerobic glycolysis a major component of this complex regulatory network. Lactate, the signature molecule of aerobic glycolysis, is a pro-inflammatory metabolite regulating cytokines and macrophage polarization [13, 18]. Lactate is generated from pyruvate by the enzyme lactate dehydrogenase (LDH). The availability of pyruvate for lactate production is regulated by pyruvate dehydrogenase (PDH), a key enzyme in the tricarboxylic acid cycle (TCA) transforming pyruvate into acetyl-CoA and subsequent mitochondrial respiration [10, 18, 19]. Interestingly, PDH quantity and activity are decreased during sepsis with resultant accumulation of pyruvate [9]. Because lactate dehydrogenase (LDH) is an equilibrium enzyme, increased lactate production during sepsis would be due to a mass-action effect exerted by an increased availability of pyruvate [20]. Consequently, a decrease in PDH activity during sepsis will result in increased lactate production and decreased mitochondrial oxidative phosphorylation [13, 21].

The PDH complex is at a key control point of energy metabolism and subject to regulation by multiple mechanisms, including succinylation of PKM2, posttranslational phosphorylation and subsequent inactivation by pyruvate dehydrogenase kinase (PDK) [22]. Dichloroacetate (DCA), a classic PDK inhibitor, has been successfully used to decrease levels of lactate and shifts metabolism towards oxidative phosphorylation in patients with congenital hyperlactatemia by directly decreasing the PDK activity with subsequent increase in the downstream enzyme, PDH, activity [23].

In the current study, we hypothesized that increased aerobic glycolysis leading to unresolved inflammation contributes to long-term complications of sepsis, including shortened lifespan [5]. We tested our hypothesis in a *D. melanogaster* model of surviving sepsis by manipulating PDH activity with DCA to modify the increased glycolysis, to normalize antimicrobial peptide expression, and improve lifespan [24, 25].

## Materials and methods

### Drosophila melanogaster strain and maintenance

The flies were raised on standard cornmeal-molasses agar medium at 22–25°C, 60% humidity, and on a 12-h light/dark cycle. The vials were changed every 3 days. We selected male flies 2–3 days after eclosion for experiments. Wild-type (WT) Canton S flies were obtained from Bloomington stock (Bloomington Drosophila Stock Center, Indiana University).

### Experimental design: Fly infection and treatment

We used our previously established model of percutaneous infection in *D. melanogaster* with subsequent antibiotic treatment to mimic the clinical course and treatment of human sepsis

[26]. The flies were divided into 3 groups: *unmanipulated*, *sham*, and *sepsis survivors*. To study the impact of DCA in diet, each experimental group was further divided into 2 groups depending on DCA treatment.

We prepared *Staphylococcus aureus* suspension in Luria-Bertani (LB) broth and grew the bacteria to an optical density (OD) of 1.0 at 600 nm. At OD of 1.0, there were ~$1.67 \times 10^6$ CFU of *S. aureus*. We anesthetized flies with $CO_2$ and pricked with a tungsten needle (0.01 mm at the tip and 0.25 mm across the needle body) into their thorax. The *sham* group was pricked with a sterile needle while the *sepsis survivor* group had the needle dipped into the bacterial solution. We pricked the sham flies first to coat the tungsten needle with hemolymph to achieve consistent bacterial coating with the infection group.

We transferred *unmanipulated*, *sham*, and *sepsis survivors* to vials containing antibiotic (linezolid) with DCA [DCA(+)] or without DCA [DCA(-)] after recovery from anesthesia and kept flies in these vials for 18 h before transferring the flies back to antibiotic-free vials. All flies were treated with linezolid (0.5 mg/mL) in the diet as described. DCA treatment continued for one week after the initial infection and it was incorporated into the diet (DCA, 0.5 mg/mL, Sigma Aldrich, St. Louis, USA) [27]. Immune and metabolic outcomes and lifespan were followed over a 7-day course.

## Fly survival and lifespan

After infection and treatment with antibiotics, fly survival was assessed by visual inspection of living flies. The flies that died within 6 h after the initial inoculation were excluded from the survival analysis, because death within the first 6 hours was considered to be secondary to trauma from inoculation rather than infection. During lifespan observations, the fly media was changed and survival assessed every 3 days. The infections were performed around the same time of the day (Fig 1).

## Rapid Iterative Negative Geotaxis (RING)

We performed geotaxis assays to evaluate the fitness-related locomotion traits following sepsis. We joined two empty polystyrene vials by tape vertically facing each other forming an

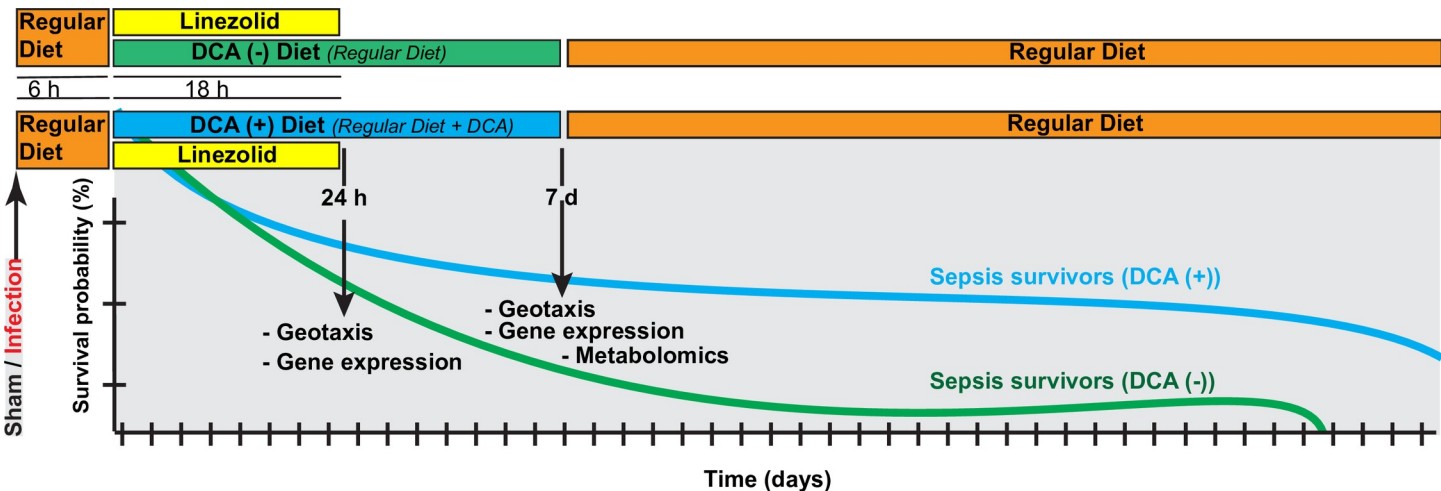

**Fig 1. Experimental design.** Conceptual depiction of the experimental design. The flies underwent either sterile needle injury ("sham") or injury with a bacteria-laden needle ("sepsis"). Following a 6-hour recovery period in vials with regular diet to assure that the needle injury did not lead to death, flies were transferred into vials containing linezolid +/- dichloroacetate (DCA). We followed the lifespan of the flies and in a subset, we measured the gene expression, geotaxis, and metabolomic changes.

18.5-cm-long tube. We transferred groups of 20 flies into the vials and allowed to acclimatize to the new setting for 5 min before conducting the assay. Flies were gently tapped down to the bottom of the vial for 10 s with the same interval and strength by the same operator throughout the whole experiment. Pictures of the flies were taken with a digital camera at 5 s. We repeated each geotaxis experiment six times, allowing for 1-min rest periods between each trial and pictures were analyzed by counting the number of flies that climb above the 10-cm mark in 5 s after the tap. We calculated the average of the number of flies crossing the 10-cm threshold and expressed the results as percentage of the total number of flies in the tube (% climbing index). Each geotaxis experiment was performed 1 h before the needle pricking (0 h baseline), 1 day, and 7 days after needle pricking. The data are presented as percent of the baseline at time 0 h.

## Patterns of host response gene expression

We determined pathogen recognition receptor Toll and antimicrobial peptide (AMP) expression with quantitative real-time polymerase chain reaction (qRT-PCR) at 24 h and 1 week after inoculation with bacteria. AMP expression levels provide an indication of the degree to which the immune system is activated, a surrogate for inflammation. Gene expression of *Toll (FBgn0262473)*, *defensin (FBgn0010385)*, *drosomycin (FBgn0283461)*, and *cecropin A (FBgn0000276)* were normalized to *actin5C (FBgn0000042)*; all data were normalized to the unmanipulated group. We used 20 flies per group in triplicate. TaqMan® Gene Expression Assay primers included the following: *toll*, Dm02151201_g1; *defensin*, Dm01818074_s1; *drosomycin*, Dm01822006_s1; *cecropin A*, Dm02609400_sH; and *actin5C*, Dm02361909_s1. Assay details are provided as (S1 Table).

## Metabolomic analysis

**LC-MS sample preparation.** Each sample containing 30 flies was homogenized in MeOH:ACN:$H_2O$ (2:2:1), snap frozen and sonicated. To precipitate the proteins, samples were centrifuged at 1500 g for 10 min at 4°C and the supernatant was transferred to a new vial and dried under $N_2$. The sample was re-suspended in 500 μL $CHCl_3$:MeOH:$H_2O$ (2:1:1) and centrifuged for 5 min at 4°C at 1500 g to separate the upper polar phase from the lower organic phase. A similar extraction was also performed for 5 mg $^{13}C$ algal cells, whose supernatant was used as an internal standard.

**Targeted analysis of organic acids.** Organic acids were analyzed by derivatizing them to their corresponding 3-nitrophenylhydrazones [28]. Briefly 100 μL of the supernatant along with 10 μL of internal standard was heated at 50°C with 50 μL of 200 mM 3-nitrophenylhydrazine in 50% aqueous acetonitrile and 50 μL of 120 mM N-(3-dimethylaminopropyl)-N-ethylcarbodiimide HCl in a 6% pyridine solution in the same solvent for 20 min. From this reaction, 100 μL was diluted to 500 μL using 50% aqueous acetonitrile (ACN). The reconstituted sample, 10 μL was injected into the Ultimate 3000 RSLC system (Thermo Fisher Scientific, Waltham, MA) connected to a Thermo Fisher Scientific Q Exactive mass spectrometer. Chromatographic separation was conducted using a reversed phase Phenomenex (Torrence, CA) Luna C18 column (2.1 mm × 150 mm, 5 μm) column using gradient elution with a mobile phase consisting of $H_2O$ + 0.1% formic acid (A) and acetonitrile + 0.1% formic acid (B), delivered at a flow rate of 0.2 mL/min. The gradient program was 3% B (0–3 min), from 3% B to 95% B (3–50 min), 95% B (50–55 min), from 95% B to 3% B (55–57 min) and 3% B (57–60 min). The mass spectrometer was equipped with an ESI source and was operated in negative ion mode using a full scan range of 150 *m/z* to 900 *m/z*. Analyte identification was confirmed

by high resolution accurate mass and compared to the $^{13}$C internal standard spectra and retention time [29] (S2 Table).

**Untargeted analysis of primary metabolites.** The primary metabolites in the polar phase were separated using ion-pairing reversed phase chromatography on the LC-MS system described above [30]. The mobile phase consisted of 5mM hexylamine in $H_2O$ (A) acetonitrile (B), delivered at a flow rate of 0.15 mL/min with a post column addition of 0.1 mL/min acetonitrile before entering the mass spectrometer. The gradient was held at 3% B for the first 3 min and increased to 30% B (3–30 min) following an additional increase to 95% B (30–55 min), a 5 min wash at 95% B (55–60 min), 95% B-100% B (60–61 min), 100% B (61–66 min), and return to initial conditions for a 3 min equilibration. The mass spectrometer was operated in negative ion mode using a full scan range of 100 *m/z* to 900 *m/z*. Analyte identification was confirmed by high resolution accurate mass and compared to the $^{13}$C internal standard. Principal component analysis (PCA) was applied to visualize grouping patterns using unsupervised multivariate data analysis.

## Statistical analysis

Kaplan-Meyer survival analysis was performed using Stata 15 (StataCorp. LLC, College Station, TX). Log-rank test was performed with the Kaplan-Meier survival curves for the groups adjusting for clusters, where each vial was treated as a cluster. Statistical analysis between different groups was accomplished with ANOVA using Graph-Pad Prism software version 8.0 (GraphPad Software Inc., La Jolla, CA).

SIMCA 14.1 (SIMCA, Umetrics, Umeå, Sweden) was used for LC-MS and Principal component analysis (PCA). Data obtained from LC-MS analysis were imported into SIMCA for multivariate analysis. PCA were carried out to discriminate the metabolic patterns between groups according to common variability within groups (unmanipulated vs. sham vs. sepsis survivors as well as diet effects among sepsis survivors) after mean centering and unit variance scaling. In our PCA analyses, R2X represents percentage variability of the X data (metabolites assessment by LC-MS raw values) explained by each principal component. The results are given as score plots of the first two principal components with their R2X values.

## Results

### DCA improves lifespan after surviving sepsis

The lifespan of flies was followed every day until death occurred in all the experimental groups of flies. The DCA treatment did not affect lifespan in *unmanipulated* and *sham* groups (S1 Fig). Interestingly, one week of DCA treatment in the *sepsis survivor* group led to significantly longer lifespan when compared to flies surviving sepsis not treated with DCA (p<0.001) (Fig 2). In sepsis survivors on regular diet, median lifespan was 12 days (IQR: 7–35 days) compared to sepsis survivors on DCA diet with a median survival of 20 days (IQR: 11–58 days). The sepsis survivors on DCA diet showed better survival throughout of the study period, especially after day 12.

### Geotaxis among sepsis survivors is not improved by DCA

The geotaxis was assessed 24-hour and 7-days after surviving sepsis. While we observed a significant lifespan benefit with DCA, there was no early separation of lifespan by day 7. There was also no improvement in geotaxis in sepsis survivors exposed to DCA compared to regular diet (Fig 3).

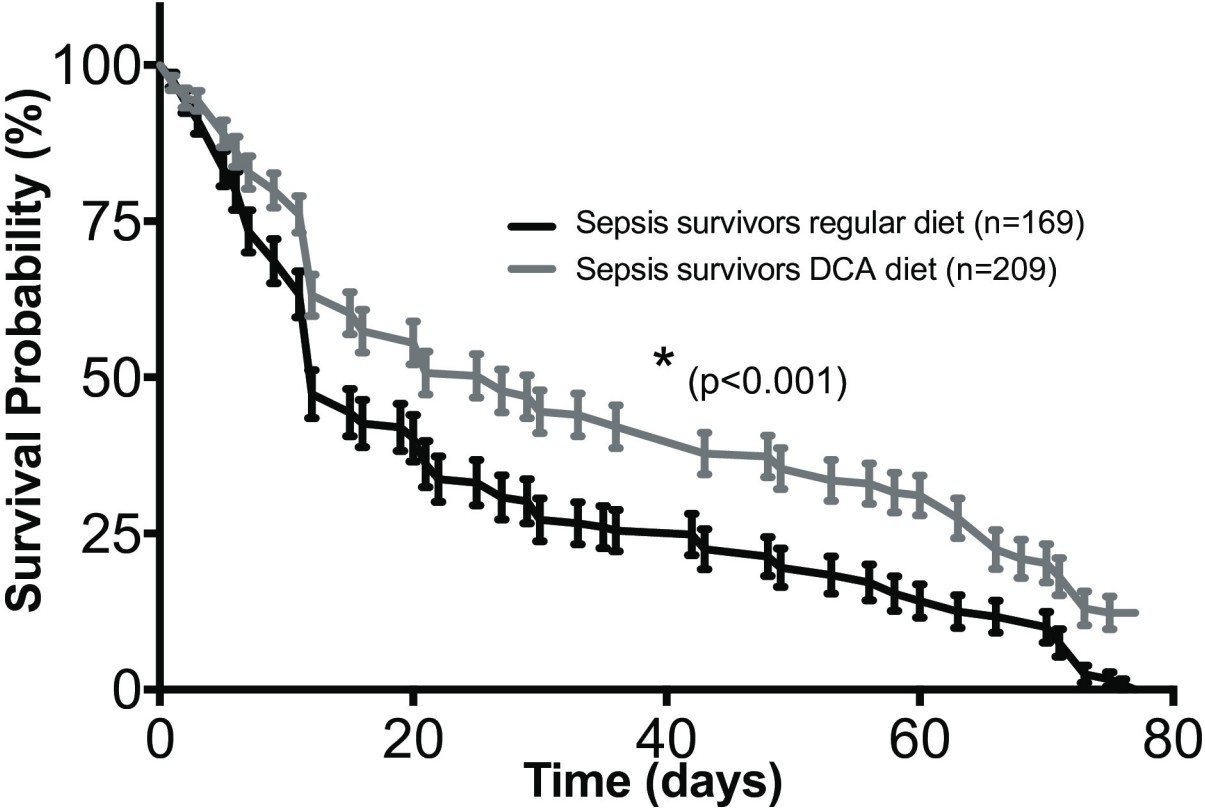

**Fig 2. Drosophila lifespan after surviving sepsis is prolonged after a 1-week exposure to DCA.** The flies were divided into 3 experimental groups: *unmanipulated*, *sham*, and *sepsis survivors*. To study the impact of DCA in diet, each experimental group was further divided into 2 groups depending on DCA treatment [DCA (-) and DCA (+)]. Survival of *Drosophila melanogaster* after septic injury with *Staphylococcus aureus* was assessed after the initial 4–6 hours to exclude trauma-associated mortality. All flies received oral linezolid (0.5 mg/mL) for 18 h. Lifespan analysis was performed using the Kaplan-Meyer survival analysis. In the DCA (+) groups, flies were fed DCA (0.5 mg/mL) only for 1 week following sepsis and then switched back to regular diet. The DCA (+) sepsis survivors had improved lifespan compared to flies that were fed regular diet (*$p < 0.001$).

## Flies treated with DCA had decreased levels of antimicrobial peptides

Gram (+) bacterial Lys-type peptidoglycan, a characteristics of Gr (+) bacteria, strongly stimulate the Toll system with downstream activation of pathways leading to AMP expression. AMP expression levels provide a surrogate marker of immune system activation [26].

We assessed the *Toll* receptor *(FBgn0262473)* along with three effector AMPs; *defensin (FBgn0010385)*, *drosomycin (FBgn0283461)*, and *cecropin A (FBgn0000276)*, all are most strongly regulated by the Toll pathway. In a recent model of Drosophila surviving sepsis, AMPs increased and remained elevated following clearance of bacterial pathogen [26]. The values reported in Fig 4 are relative values to the one measured in the unmanipulated groups [DCA (-) and DCA (+)] at 24 hr and 1-week post sepsis (Fig 4).

In the early post-sepsis phase (24 h), the *cecropin A (FBgn0000276)* expression was significantly higher among flies surviving sepsis on both regular (10.2-fold) and DCA (22.6-fold) diet. The expression of *defensin (FBgn0010385)* was significantly increased among sepsis survivors exposed to DCA 24 h after septic injury compared to sham flies fed DCA (17.9-fold). *Defensin (FBgn0010385)* was also significantly elevated in the DCA (+) sepsis survivors compared to DCA (-) sepsis survivors (11.4-fold). The transmembrane protein *Toll (FBgn0262473)* expression was not affected by the DCA treatment.

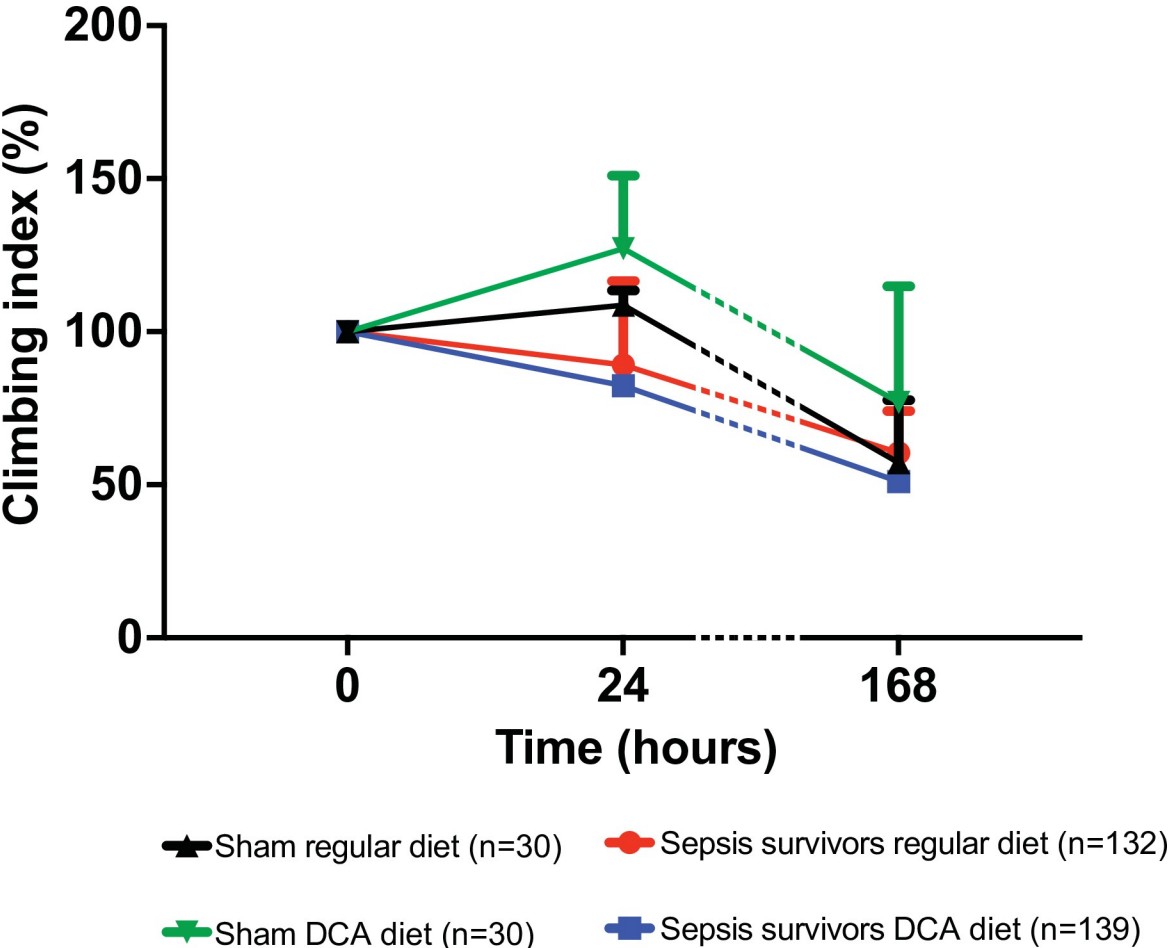

**Fig 3. Geotaxis among sepsis survivors is not improved by DCA.** Despite lifespan advantage provided by the 1-week exposure to DCA, geotaxis–a composite index of locomotor performance- did not improve negative geotaxis among survivors of sepsis.

When the same set of genes were examined 1 week after the injury, expression of *drosomycin (FBgn0283461)* (29-fold in DCA (-) and 25.4-fold in DCA (+)) and *cecropin A (FBgn0000276)* (3.11-fold in DCA (-) and 4.8-fold in DCA (+)) were significantly increased in sepsis groups when compared to sham groups regardless of the diet. However, DCA treatment significantly decreased the expression of *drosomycin (FBgn0283461)* (0.43-fold) and *cecropin A (FBgn0000276)* (0.75-fold) in flies surviving sepsis.

While DCA treatment significantly diminished *drosomycin (FBgn0283461)* and *cecropin A (FBgn0000276)* expression, the expression of *defensin (FBgn0010385)* was significantly higher in DCA-treated flies after sepsis compared to sepsis survivors on regular diet (2.4-fold). In summary, DCA treatment for 1 week facilitated a focused and sustained expression of *defensin (FBgn0010385)*, the AMP against gram-positive bacteria, whereas DCA treatment reduced expression of *drosomycin (FBgn0283461)* and *cecropin A (FBgn0000276)*.

## Treatment of Drosophila surviving sepsis with DCA demonstrates metabolomic reprogramming

We then studied changes in metabolite levels in *sham* and *sepsis survivor* groups with or without DCA treatment. We used multivariate analysis with an unsupervised PCA model to study

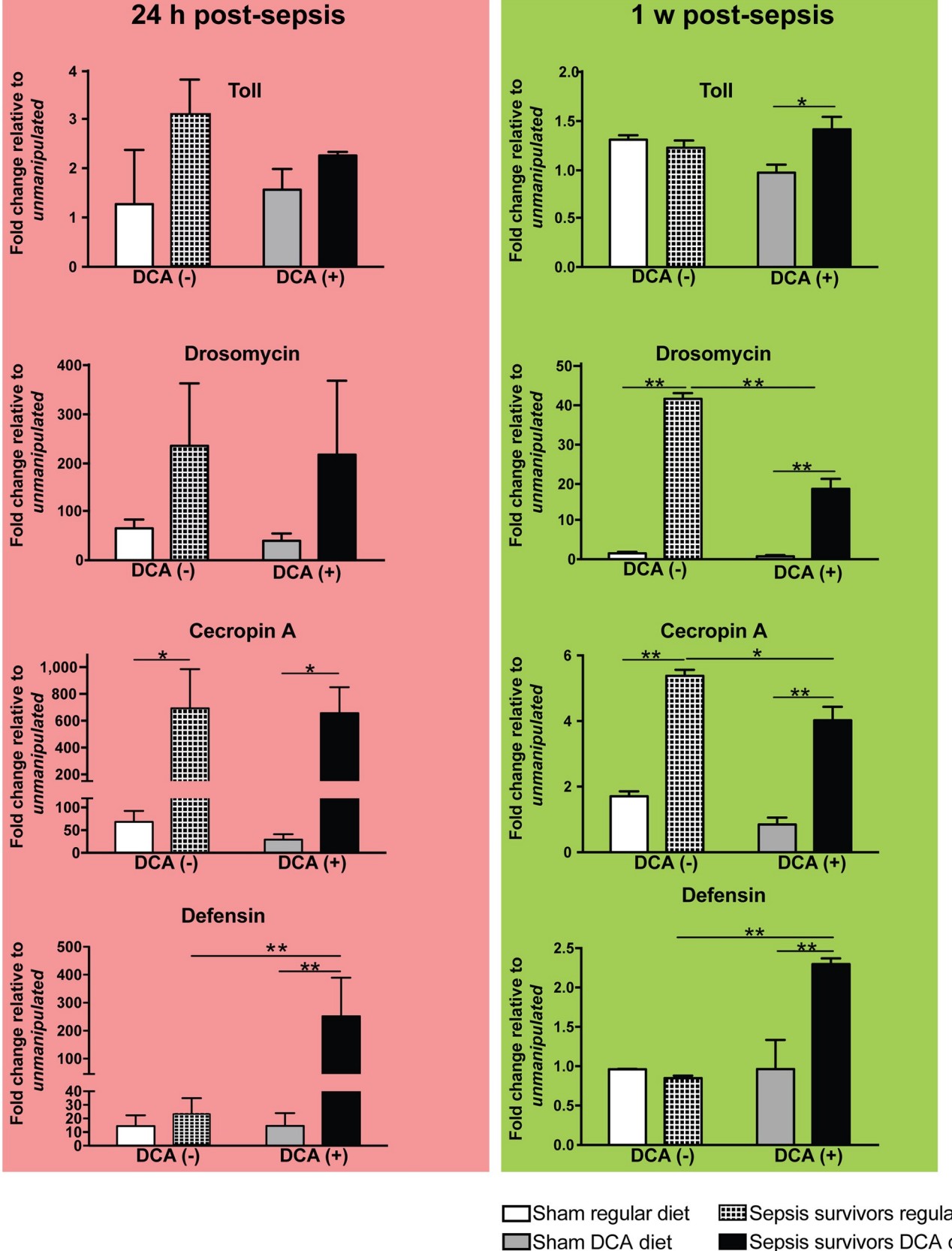

**Fig 4. DCA selectively affects early and late AMP expression patterns in Drosophila surviving sepsis.** Flies were infected with Staphylococcus aureus and then treated with orally available linezolid (0.5 mg/mL) in the diet for 18 hours. DCA treatment continued for one week after the initial infection and it was also incorporated into the diet (DCA, 0.5 mg/mL). AMP transcription was assessed after 24 h and 1 week surviving sepsis by quantitative PCR and normalized to unmanipulated flies in each respective diet group. Significant different expression values across treatment within each antimicrobial peptide transcript were determined by one-way ANOVA and Tukey's multiple comparison tests ($^*<0.05$ and $^{**}<0.01$). After 24 h surviving sepsis, *cecropin A* expression was significantly elevated regardless of the diet exposure. However, only the DCA exposed flies had significantly elevated *defensin* expression levels. After 1 week surviving sepsis, addition of DCA significantly decreased the expression of *cecropin A* and *drosomycin* while increasing *defensin* expression. The number of biological repeats for qPCR was n = 6.

metabolites. PCA results were calculated by reducing the number of dimension while preserving the initial information (*metabolites variability*) (S2 Table) [31]. The two principal components of the score plot are considered as a combination of the initial variables, each of them having a different "weight" in the calculation of the principal component (PC). The weights of each variable on the PC1 and PC2 describe the metabolites variation responsible for the discrimination of the clusters in the PCA score-plots. The multivariate analysis with the unsupervised PCA model showed that flies on regular diet could not be separated according to their group (Fig 5A). On the contrary, when flies were treated with DCA, the PCA score plot shows that the 2 first components could discriminate sham from unmanipulated and from sepsis survivors (Fig 5B). When comparing the DCA effect among the sepsis survivors, the PCA model could clearly discriminate two clusters along the first component (R2X = 0.885) (Fig 5C). The metabolites having the highest weight on the first component are α-ketoglutarate, fumarate, and pyruvate. These PCA models confirm the metabolic consequences of the DCA treatment. For the further comparison of metabolic assessment, we used the unmanipulated flies to normalize metabolite concentrations in *sham* and *sepsis survivor* groups.

From the results shown in Fig 6, two metabolomic effects became obvious. The first observed difference was the effect of infection. We observed aerobic glycolysis with acetate, lactate, and TCA metabolite accumulation such as pyruvate, citrate/isocitrate, α-ketoglutarate, succinate, and fumarate among *sepsis survivors* on regular diet compared to *sham* groups on regular diet. The second observation was the diet effect. DCA treatment has no effect on sham flies as none of the metabolites measured were significantly modified with the introduction of DCA. Once we introduced DCA into the diet, the infection effect was partially reversed. DCA (+) resulted in a decrease in lactate levels among sepsis survivors, with a return to levels similar to that in the sham group. Only malate and fumarate were significantly lower in *sepsis survivors* compared to *sham* group while on DCA. In addition, DCA diet led to decreased fumarate, α-ketoglutarate and pyruvate levels in comparison to sepsis surviving flies on regular diet. Among sepsis survivors, levels of pyruvate, citrate, α-ketoglutarate, acetate, and succinate were normalized to the levels of sham in the DCA group (Fig 6).

Levels of acetyl-CoA in DCA (+) flies were significantly higher when compared to flies on regular diet for each corresponding group (p<0.05). However, neither in regular diet, nor in the DCA diet groups, there were no differences between the *sham* and *sepsis survivor* groups within each respective group (Fig 6).

## Discussion

Sepsis was recently defined by the SEPSIS 3 consortium as a 'life-threatening organ dysfunction caused by a dysregulated host response to infection' [24]. Acute inflammation during early phases of sepsis is necessary to survive infection, yet sustained inflammation is deleterious [26, 32, 33]. Activated immune cells during early phases of sepsis undergo rapid activation to generate ATP and biosynthetic intermediates utilizing aerobic glycolysis and lactate accumulation, which in turn propagates further antibacterial defenses and inflammation [13, 18,

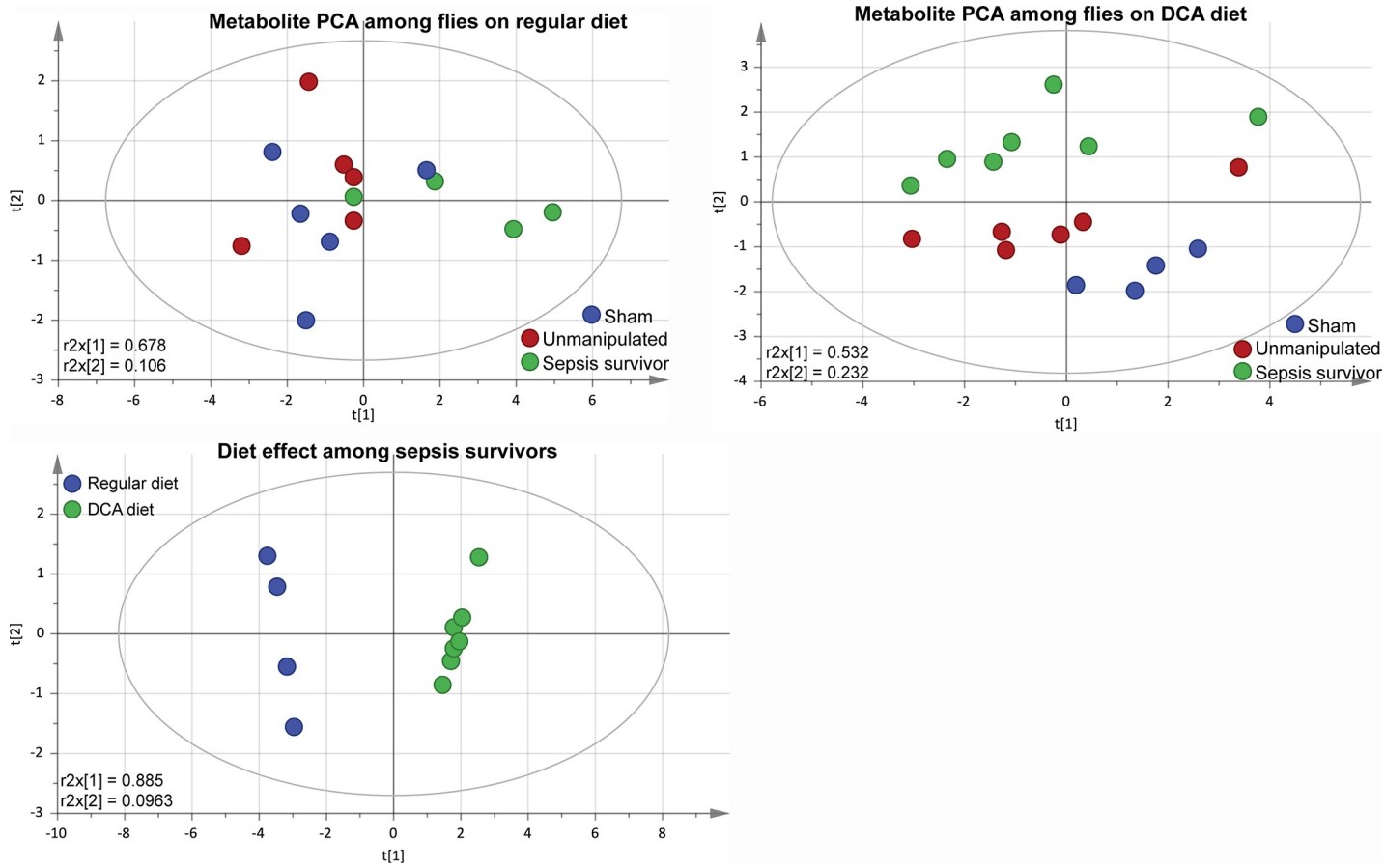

**Fig 5. Multivariate analysis of metabolomic LC-MS data show DCA-dependent changes among sepsis survivors.** We studied changes in metabolite levels in *sham* and *sepsis survivors* with or without DCA treatment. We used the *unmanipulated* flies to normalize metabolite concentrations in *sham* and *sepsis survivor* groups. (A) Multivariate analysis with unsupervised PCA model showed that flies on regular diet could not be separated according to their group (r2x = 0.678) (**Fig 5A**). (B) However, when flies were treated with DCA, PCA score plot shows that the 2 first components could discriminate *sham* from *unmanipulated* and *sepsis survivors* (r2x = 0.532) (**Fig 5B**). (C) Finally, when we compared the effects of DCA among the sepsis survivors, the PCA model could clearly discriminate two clusters along the first component (r2x = 0.885) (**Fig 5C**). The metabolites having the highest weight on the first component were α-ketoglutarate, fumarate, and pyruvate. The number of biological repeats for LC-MS was n = 5.

34]. In sepsis survivors, such an excessive acute immune response could transition to dysfunctional immunity with long-term poor outcomes [18, 19]. As such, the new paradigm in the management of sepsis is towards improving potentially modifiable variables of late morbidity and mortality [35–37]. However, causality between sepsis and "*post-acute mortality*" in sepsis survivors is not well established [38, 39]. Metabolic profiling of rodents and humans show a shift from oxidative phosphorylation to aerobic glycolysis with increased lactate production during early phases of sepsis [10, 40, 41]. While metabolic changes during an acute immune response are not novel, recent work conceptualized that acute inflammatory responses in immune cells are both supported and regulated by metabolic shifts leading to "trained immunity" in innate immune cells [18, 42]. In the current work, we used the *Drosophila* sepsis survival model to understand the association between innate immunity and metabolomic changes during long-term sepsis sequelae. Drosophila provides the advantage that it only has the innate immunity, yet survives infections and maintains an immune memory (*trained immunity*) as a whole organism [43, 44]. Trained immune memory confers a long-term protection against secondary infections, yet may consume resources for the organism to live and

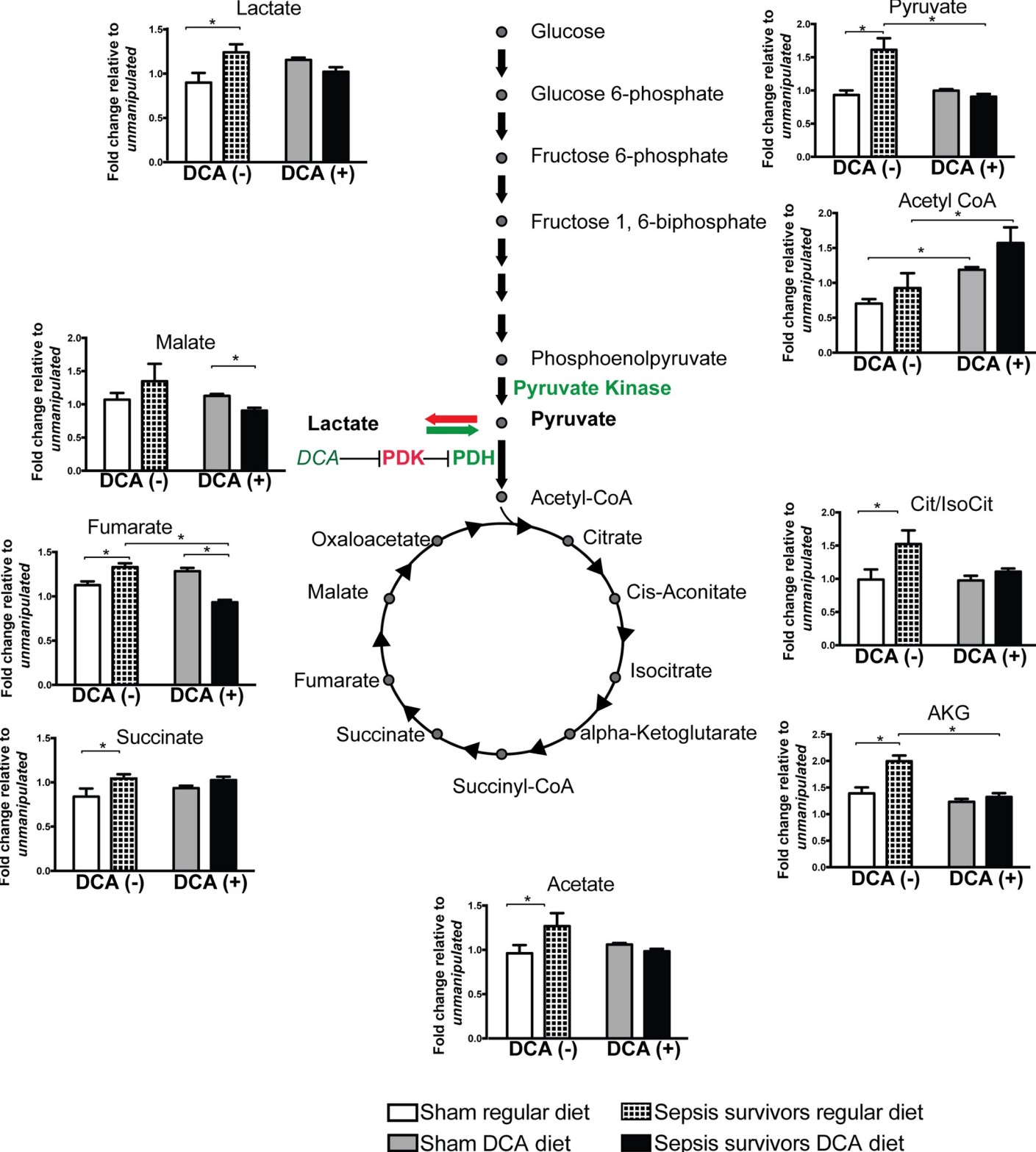

**Fig 6. DCA promotes a shift from aerobic glycolysis to oxidative phosphorylation in flies surviving sepsis.** We studied changes in metabolite levels in *sham* and *sepsis survivors* with or without DCA treatment. We used the *unmanipulated* flies to normalize metabolite concentrations in *sham* and *sepsis survivor* groups. Sepsis led to acetate, lactate, and TCA metabolite accumulation such as pyruvate, citrate/isocitrate, α-ketoglutarate, succinate, and fumarate among *sepsis survivors* on regular diet compared to *sham* groups on regular diet. When flies were fed DCA, infection effect was partially reversed with a decrease in lactate among sepsis survivors returning to

levels similar to that in the sham group. Only malate and fumarate were significantly lower in *sepsis survivors* compared to *sham* group while on DCA. In addition, DCA diet led to decreased fumarate, α-ketoglutarate and pyruvate levels in comparison to sepsis surviving flies on regular diet. Once exposed to DCA, the TCA metabolites were either brought back to baseline–like succinate- or decreased lower than sham in sepsis survivors. In summary, DCA reversed the systemic metabolic signatures of aerobic glycolysis among survivors of sepsis. The number of biological repeats for TCA analysis was n = 5.

survive on the long run, a pragmatic trade-off [7, 45–50]. This becomes especially relevant in organisms that lack adaptive immune systems, such as the Drosophila. Key metabolic enzymes of glucose oxidation and citric acid cycle are studied within the context of *trained immunity* [19, 51]. One of the upstream enzymes in glucose oxidation, pyruvate dehydrogenase (PDH), is a key regulatory enzyme determining the fate of pyruvate into the TCA cycle and its activity is decreased in sepsis [9, 52].

While acknowledging the limitations of pre-clinical models, others and we defined "sepsis" in Drosophila, where the flies were infected with Staphylococcus and then treated with orally available antibiotics [25]. The antibiotic exposure eliminated the bacterial burden, however inflammation ("dysregulated host response") persisted. In our original work, we showed decrease and subsequent elimination of bacterial burden with antibiotic treatment.

We tested our hypothesis that manipulating PDH activity with a small molecular PDH activator, DCA, will reverse the increased glycolysis in sepsis, normalize AMP expression, and improve lifespan. Pyruvate dehydrogenase kinase 1 (PDK1) is upstream of PDH, phosphorylates and inhibits PDH preventing conversion of pyruvate to acetyl-CoA and shifting cell metabolism towards aerobic glycolysis highlighted by increased lactate production. DCA, by inhibiting PDK1, activates PDH and promotes the entry of pyruvate into the TCA cycle [53]. We here showed that 1-week exposure to DCA improves lifespan of Drosophila surviving sepsis over a course of almost 12 weeks, regulates inflammatory AMP, and promotes conversion of pyruvate into acetyl-CoA facilitating oxidative phosphorylation over aerobic glycolysis. DCA is reported to promote lifespan extension in *C. elegans* and *D. melanogaster* upon continuous exposure, yet we did not observe such an effect in the control flies exposed to DCA as we exposed flies for only 1 week and subsequently reared them on regular diet [54, 55]. The relatively brief exposure to DCA could explain the lack of survival advantage in non-infected control flies. Nevertheless, DCA effect on lifespan among flies surviving sepsis was profound and did last beyond the 1-week exposure. DCA has ben used to manipulate metabolic and inflammatory changes in other infection models. Work by Yamane et al did show in a murine model of influenza that both DCA as well as a DCA analog (diisopropylamine dichloroacetate) improved survival associated with inhibition of pyruvate dehydrogenase kinase 4 [56]. While we observed increased lifespan, we did not observe a functional recovery among sepsis survivors as evaluated with geotaxis [56, 57].

Fungi or gram-positive bacteria activate the Toll pathway with subsequent activation of the NF-κB factor Dif, Relish, or Dorsal and production of AMPs such as *drosomycin*, *cecropin A*, and *defensin* [58–61]. We studied AMP's as a surrogate for inflammation based on our earlier research [26]. In our prior work using the same model, we explored the components of the canonical transcription factor NF-κB (dorsal, Dif, Relish) and a wider panel of receptors, signaling molecules, and antimicrobial peptides (PGRP-SD, Toll, Metchnikowin, Cecropin A, JNK, Drosomycin, Defensin, InR, IRS, PTEN, Akt1, Foxo, mTORC1, and ratio of p-Akt/Akt) as surrogates for sustained inflammation among flies surviving sepsis. We observed persistent elevation of NF-κB gene expression as well as activation and AMP's in the absence of any obvious infection as shown by culturing flies surviving sepsis. This enabled us to establish the basis for the "sepsis survivor" phenotype with the goal of mimicking patients surviving sepsis, yet having ongoing inflammation even at hospital discharge.

In our experiments, among flies not treated with DCA, the *drosomycin* and *cecropin A* expression was elevated 1 week after surviving sepsis despite clearance of bacterial burden suggesting a sustained expression and activation of AMPs. Activation of AMPs, markers of inflammation, could contribute to lifespan reduction. Support for this idea comes from the observations by the Ganetzky laboratory that overexpression of AMPs, such as *drosocin*, *attacin*, *defensin* and *drosomycin*, in young flies induces neurodegeneration in mature flies and shortens lifespan [32]. Similarly, AMP overexpression in flies deficient in *Methuselah-like receptor-10* (*Mthl10*) did limit lifespan in Drosophila [50].

The metabolite patterns of flies surviving sepsis reared on regular diet showed characteristics of the aerobic glycolysis with lactate and TCA cycle metabolites accumulating. Lactate has been used as a marker for poor clinical outcome and, in our experience, high lactate level in Drosophila is associated with a shortened lifespan [19, 26, 62]. In addition to lactate accumulation among sepsis survivors, pyruvate, citrate/isocitrate, α-ketoglutarate, acetate, succinate, and fumarate were significantly elevated compared to sham group. However, following DCA treatment, lactate and pyruvate levels came to baseline, suggesting a shift of pyruvate into the TCA cycle away from the lactate production. Among TCA cycle metabolites, citrate/isocitrate, α-ketoglutarate, acetate and succinate also came to baseline levels with DCA treatment. Fumarate and malate levels decreased significantly in the DCA group. The decrease in fumarate and malate may also indicate a redirection of flux from pyruvate carboxylase to PDH entry of pyruvate, yet this requires flux studies [52]. We used unsupervised PCA method to analyze the metabolomic changes. When we studied the metabolic impact of DCA treatment in the sham group, DCA diet has no metabolic effects compared to regular diet other than an increase in acetyl-CoA. However, in the sepsis survivor group, DCA diet effects on metabolite contents are in good agreement with the DCA mechanism of action and the expected reversal of aerobic glycolysis. Interestingly, when compared to the sham group on regular diet, the DCA-treated sepsis-surviving group, none of the metabolites differed between these two groups. This result may support our hypothesis that metabolomic effects of sepsis have been reversed by DCA. When regular diet- and DCA- survivors were compared, we observed a clear clustering in the PCA loading plots.

How the relatively short course of DCA affects the sustained levels of anti-Gr(+) *defensin* expression is of interest as it could link metabolomic changes to inflammation. Pyruvate can be converted to acetyl-CoA in the nucleus by the nuclear PDH, providing a source of acetyl for histone acetylation and DCA could act in a similar fashion promoting histone acetylation and thus linking metabolic changes with epigenetic control of AMP expression [63–66].

We studied geotaxis up to day 7 to avoid any survivor bias effect on later time points and found no difference between the regular and DCA diet groups on day 1 and day 7 after sepsis. Our goal of using a functional outcome was to mimic the human experience surviving sepsis [57].

As for the limitations of our current work, the DCA effect on the life span is not mechanistically shown and requires cellular level genetically modified flies to test our hypothesis further along with $^{13}$C-based metabolomic flux studies.

In summary, our results suggest an association between DCA-induced metabolic changes in Drosophila surviving sepsis and their lifespan. This sepsis survival model in an organism with only innate immunity lends itself for further mechanistic exploration of various spatial and temporal interventions towards lifespan and healthspan outcomes.

## Supporting information

**S1 Fig. Drosophila lifespan after sterile needle injury ("sham") is unchanged following a 1-week exposure to regular diet or DCA.** To study the impact of DCA in diet, sterile needle

injured ("sham") flies were divided to either receive regular or DCA diet. Survival of *Drosophila melanogaster* after sterile injury was assessed following the initial 4–6 hours to exclude trauma-associated mortality. All flies received oral linezolid (0.5 mg/mL) for 18 h. Lifespan analysis was performed using the Kaplan-Meyer survival analysis. In the DCA diet group, sham flies were fed DCA (0.5 mg/mL) only for 1 week following needle injury and then switched back to regular diet. There was no lifespan difference between regular and DCA diet receiving sham flies ($p > 0.05$).
(TIF)

**S1 File. Data for publication.** This folder contains all of the data for the manuscript and the data files are titled with their corresponding figure panel.
(ZIP)

**S1 Table. RT-qPCR settings.**
(DOCX)

**S2 Table. LC-MS settings.**
(DOCX)

## Author Contributions

**Conceptualization:** Ata Murat Kaynar.

**Data curation:** Veli Bakalov, Laura Reyes-Uribe, Rahul Deshpande, Abigail L. Maloy, Stacy Gelhaus Wendell, Ata Murat Kaynar.

**Formal analysis:** Veli Bakalov, Derek C. Angus, Chung-Chou H. Chang, Laurence Le Moyec, Stacy Gelhaus Wendell, Ata Murat Kaynar.

**Investigation:** Laura Reyes-Uribe.

**Methodology:** Veli Bakalov, Laura Reyes-Uribe, Ata Murat Kaynar.

**Writing – original draft:** Veli Bakalov, Laura Reyes-Uribe, Abigail L. Maloy, Steven D. Shapiro, Derek C. Angus, Laurence Le Moyec, Stacy Gelhaus Wendell, Ata Murat Kaynar.

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
