## [Decision Letter · Decision Letter 0]

25 Mar 2020

PONE-D-20-05383

Reversal of Warburg effect via metabolic reprogramming improves lifespan in a Drosophila model of surviving sepsis.

PLOS ONE

Dear Dr. Kaynar,

Thank you for submitting your manuscript to PLOS ONE. After careful consideration, we feel that it has merit but does not fully meet PLOS ONE’s publication criteria as it currently stands. Therefore, we invite you to submit a revised version of the manuscript that addresses the points raised during the review process.

Please pay attention to both the pitfalls of overinterpretation, but also the requests to repeat experiments where the numbers of flies is limiting. "Is the manuscript technically sound, and do the data support the conclusions?" The answer to this question needs to be affirmative for the work to be considered for publication. I note that both reviewers think the question posed is a good one and I encourage the authors to engage with the considered criticisms.

We would appreciate receiving your revised manuscript by May 09 2020 11:59PM. To enhance the reproducibility of your results, we recommend that if applicable you deposit your laboratory protocols in protocols.io, where a protocol can be assigned its own identifier (DOI) such that it can be cited independently in the future. For instructions see: http://journals.plos.org/plosone/s/submission-guidelines#loc-laboratory-protocols

We look forward to receiving your revised manuscript.

Kind regards,

Fanis Missirlis, Ph.D.

Academic Editor

PLOS ONE

Journal Requirements:

2. Thank you for stating the following financial disclosure: "N/A"

a)    Please provide an amended Funding Statement that declares *all* the funding or sources of support received during this specific study (whether external or internal to your organization) as detailed online in our guide for authors at http://journals.plos.org/plosone/s/submit-now.  

b)    Please state what role the funders took in the study.  If any authors received a salary from any of your funders, please state which authors and which funder. If the funders had no role, please state: "The funders had no role in study design, data collection and analysis, decision to publish, or preparation of the manuscript."

Reviewers' comments:

Reviewer's Responses to Questions

**Comments to the Author**

1. Is the manuscript technically sound, and do the data support the conclusions?

Reviewer #1: No

Reviewer #2: No

2. Has the statistical analysis been performed appropriately and rigorously? 

Reviewer #1: No

Reviewer #2: Yes

3. Have the authors made all data underlying the findings in their manuscript fully available?

Reviewer #1: Yes

Reviewer #2: Yes

4. Is the manuscript presented in an intelligible fashion and written in standard English?

Reviewer #1: Yes

Reviewer #2: Yes

5. Review Comments to the Author

Reviewer #1: The manuscript investigates the protective effect of inhibition of glycolysis and lactate production on lifespan reduction that is observed after sepsis. Experiments are conducted with the chemical compound DCA which is known to stimulate oxidative phosphorylation over glycolysis. One week after sepsis DCA results in clear changes in the expression of antimicrobial peptide (AMP) genes (as an indication of an activated immune system) such as a reduction in Cecropin A and Drosomycin but an increase in Defensin. Targeted metabolite analysis shows some of the expected changes following DCA treatment such as the absence of increase in lactate after sepsis in DCA-treated animals while also differences are observed in the abundance of TCA cycle intermediates.

The manuscript presents interesting data but there is a need for a better explanation and expansion of the data as well as a better description of the experiments.

Better explanation and expansion of the data:

1) The manuscript would benefit from a better outline of the experiments, such as a schematic of the timing of the treatment of the flies and at which time point samples are taken for analysis.

2) The concept of “sepsis” needs to be better introduced. Are there any data that show the extent of infection of the flies after pricking them with bacterial solution?

3) What is the specificity of DCA? Has it been used in other experiments that investigate the Warburg effect?

3) As an output of activation of the immune response (as an approximation of the concept of inflammation? – needs to be explained), only a few AMP genes and Toll are selected. This seems too limited in order to have a good idea about the status of “sepsis survivor” in the treated flies.

4) The purpose of the geotaxis experiments is not explained and should be elaborated.

5) The purpose of the experiments of untargeted metabolomics is not explained and should be elaborated. Particular metabolites have the highest weight in principal component analysis: explain and provide evidence.

6) In the re-infection experiments, unmanipulated flies are compared with sepsis survivors. But sham-treated animals should be used for comparison?

Better description of experiments:

1) The number of biological repeats should be clearly mentioned in text and figure legends. The data in the supplementary figures correspond to technical or biological repeats (e.g. qPCR)?

2) The criteria for the identification of the targeted metabolites (pyruvate, lactate, TCA intermediates) needs to be described (m/z values, retention times in chromatography…), perhaps as supplementary data.

3) Primers and conditions for qPCR need to be provided.

4) Supplementary Figure 6 needs to be explained. There are no legends for the supplementary data in general.

Reviewer #2: Review on manuscript Reversal of Warburg effect via metabolic reprogramming improves lifespan in a Drosophila model of surviving sepsis

1. Summary of the research and my overall impression

The general topic of this research article is highly entertaining. Metabolic change of activated immune cells, accompanied by the shift in their role and its impact on overall survival of acute and chronic infection, has gained a lot of attention recently and represents a potential target for medical treatment.

Authors of the manuscript focused on long-term changes induced in response to bacterial infection and claim that these changes, resembling the Warburg effect, can be reversed by feeding the flies with dichloroacetic acid, which improves the outcome of the sepsis on lifespan.

Even though I like the ideas behind the design of the experiment and building of the story, the data do not entirely adhere to their interpretation and do not support the general conclusions.

Therefore, I have to declare here that data seems to me to be over-interpreted and does not support the presented interpretations. Misinterpretation of some observations, together with a low number of individuals used in survival and lifespan assays, further supports my feeling that this manuscript has to improve a lot to reach common standards of PLOS ONE journal.

2. Discussion of specific areas for improvement

Major points:

1) It is known that bacterial infection causes polarization of macrophages towards aerobic glycolysis in drosophila. Nevertheless, all the detected changes (gene expression and metabolites) have been measured on the whole fly lysates. Additionally, the treatment affecting the metabolic rearrangement of cells towards oxidative phosphorylation is systemic. The presented data, in my opinion, do not support the metabolic shift towards aerobic glycolysis. The mechanism of the DCA effect on the life span is thus elusive and cannot be interpreted as a Warburg effect reversal. Therefore the mechanism of action of this drug is entirely unknown may influence many different signaling pathways in various tissues and organs. Moreover, Drosophila - as a model organism, enables to solve particularly the question of tissue-specific effects of such treatment due to versatile genetic tools that were omitted entirely.

For me, the whole model of survivors of sepsis and the long-term persisting Warburg effect in flies is a little bit confusing. Authors claim that flies surviving the infection still show increased activation of the immune system and metabolic reprogramming. Nevertheless, in this manuscript, there are completely missing data about the level of bacteria and how long they are persisting in the flies. While trying to find the answer to this question, I found the answer to this question in supplementary data of previously published research publication establishing this experimental model (citation 23). According to the data, there are still bacteria present in the flies, even after curing the sepsis by antibiotics. Therefore, we can expect that persisting changes are caused by this chronic bacterial infection rather than cured acute infection as it is interpreted (Link to the previously published data from the corresponding author https://static-content.springer.com/esm/art%3A10.1186%2Fs40635-016-0075-4/MediaObjects/40635_2016_75_MOESM1_ESM.pdf).

2) Many data that are presented in the manuscript do not support the conclusion because of the following reasons:

Fig.1 – the number of individuals is not sufficient to claim anything about survivors of infection; it is not clear to me whether more than one biological replicate has been done.

Fig.2 – I would say that there are not enough individuals used in this assay, and the number of individuals in compared groups is strikingly different.

Fig.3 – Several significant differences do not have displayed the significances in the plots even though they are discussed in the text. Moreover, the data show a striking difference even in sham control groups in comparison to untreated ones. How can the authors explain that the expression level of cecropin-A and drosomycin is increased more than 50-times just in response to sham treatment?

Fig.4 – I like the PCA analysis (although it would be nice to see data also in RDA), but have some problems with some interpretations. How is figure 4A, showing no difference between all three groups, support the notion that infection induces the Warburg effect? Why these groups diversify because of DCA feeding (because one would instead expect that DCA treatment would diminish previously observed changes)? Further, the variance is explained mainly by metabolites of the TCA cycle metabolites (as it is mentioned in the results), but the typical changes for the Warburg effect are instead increased glycolysis and pentose phosphate pathway while the TCA cycle undergoes complicated rewiring. Are these metabolites changing as well. How are these data in adherence with previously published observation, where for instance, lactate has not been altered in the same treatment? (https://www.mdpi.com/2218-1989/6/4?view=abstract&listby=date&page_no=1)

Fig.5 – According to my opinion, the data are not interpreted correctly since the authors claim that DCA leads to increased TCA cycle rather than the conversion of pyruvate to lactate. Still, there is almost no difference in most of the TCA metabolites in the sham group of flies in response to DCA feeding.

Fig.6 – The analysis has performed to show whether gain of primed immunity to S. aureus reinfection rely in metabolic rearrangement and can be thus disturbed by DCA feeding. Unfortunately, I have to admit that this experiment failed to show that there is the primed immunity in analyzed flies. The number of individuals in this assay is insufficient. By adding the sham manipulated group that was excluded from the figure but can be found in data files, it is clear the flies did not gain the primed immunity at all. The rest of the experiment is thus irrelevant.

Minor points:

Line4 – I do not entirely understand the last sentence of the abstract, but it may be due to my English.

Introduction – I lack in the introduction the information about what is known about the Warburg effect and polarization of macrophages in Drosophila. Moreover, many of the references are not very well supporting the notions (for example – 17, 19, etc.).

Material and Methods

The paragraphs - Fly reinfection and Fly survival and lifespan - should be described more into the detail.

Even though there is mentioned the source of primers, it would be valuable to have the sequences of primers and probes used in the appendix. Further, the genes, included in the publication, should be identified by their FLY base FBGN identifier.

Discussion – the references should be revisited. The notion on line 331-334 seems to me excessive and directly connected with the published data.

Based on the justification mentioned above, I have to say that I do not recommend the manuscript to be accepted.

In Ceske Budejovice (Czech Republic) 21-3-2020 Adam Bajgar

6. PLOS authors have the option to publish the peer review history of their article (what does this mean?). If published, this will include your full peer review and any attached files.

Reviewer #1: No

Reviewer #2: Yes: Adam Bajgar, Ph.D.

---

## [Author Response · Author response to Decision Letter 0]

28 Sep 2020

September 10th, 2020

Fanis Missirlis, Ph.D.

Academic Editor

PLOS ONE

Departamento de Fisiología, Biofísica y Neurociencias, 

CINVESTAV-IPN, Zacatenco, 07360, México

RE: PONE-D-20-05383 “Reversal of Warburg effect via metabolic reprogramming improves lifespan in a Drosophila model of surviving sepsis.”

Dear Prof. Missirlis:

During these tumultuous times with COVID-19 pandemic, we would like to thank the editorial board and reviewers for their constructive comments and time for our manuscript entitled “Reversal of Warburg effect via metabolic reprogramming improves lifespan in a Drosophila model of surviving sepsis.” We here are submitting a revised version of our work and agree with all the reviewers that overinterpretation of data should be avoided. Accordingly, we revised our discussion significantly and wrote it based on the available data. 

The repeat experiments for “trained immunity” are valuable suggestions, however during the COVID-19 pandemic we couldn’t perform new experiments and we are finally getting a chance to restart our work. We did remove the re-infection data within the “trained immunity” context without losing the message of the manuscript. We hope that our work will satisfy the review process.

Sincerely,

A. Murat Kaynar

 

Journal Requirements: 

Critique: We note that you have included the phrase “data not shown” in your manuscript. Unfortunately, this does not meet our data sharing requirements. PLOS does not permit references to inaccessible data. 

Response: We did add the survival graph for the unmanipulated and sham groups as an Supporting Information file. 

Critique: Please include captions for your Supporting Information files at the end of your manuscript, and update any in-text citations to match accordingly. 

Response: We added the captions accordingly. 

Critique (9/26/2020): 

Response: Dr. Kaynar received research grant support from NIH (HL126711). 

Response: The funders had no role in study design, data collection and analysis, decision to publish, or preparation of the manuscript.

Response: Dr. Kaynar received salary support from NIH (HL126711).

Reviewers' comments:

We thank the reviewer for constructive input and we will address these in an itemized fashion.

1. Is the manuscript technically sound, and do the data support the conclusions?

Reviewer #1: No

Reviewer #2: No

Response: We thank the reviewers and have rewritten the discussion avoiding overinterpretation of the data.

2. Has the statistical analysis been performed appropriately and rigorously?

Reviewer #1: No

Reviewer #2: Yes

Response: Our co-author, Dr. Chung-Chou H. (Joyce) Chang, is Professor of Biostatistics and Clinical and Translational Science in the University of Pittsburgh Graduate School of Public Health and we have been collaborating on various models of sepsis for more than 10 years. Individual statistical responses will follow the specific statistical questions.

 

Reviewer #1: 

Critique: The manuscript would benefit from a better outline of the experiments, such as a schematic of the timing of the treatment of the flies and at which time point samples are taken for analysis.

Response: We thank the reviewer and added a schematic for experimental outline and read-outs as Figure 1 hoping it clarifies the study. Our message with this paper is the long-term effects of a 1-week treatment with DCA. After 1-week exposure to diet with (+) or without (-) DCA, all sepsis-surviving flies were switched to regular diet. 

Critique: The concept of “sepsis” needs to be better introduced. Are there any data that show the extent of infection of the flies after pricking them with bacterial solution? 

Response: “Sepsis” was recently defined by the SEPSIS 3* consortium as a ‘life-threatening organ dysfunction caused by a dysregulated host response to infection.’ While acknowledging the limitations of pre-clinical models, others# and we@ defined “sepsis” in Drosophila, where the flies were infected with Staphylococcus and then treated with orally available antibiotics. The antibiotic exposure eliminated the bacterial burden, however inflammation (“dysregulated host response”) persisted. In our original work, we showed decrease and subsequent elimination of bacterial burden with antibiotic treatment. 

*[https://pubmed.ncbi.nlm.nih.gov/26903338 The Third International Consensus Definitions for Sepsis and Septic Shock (Sepsis-3) JAMA 2016 Feb 23;315(8):801-10.]

#[https://pubmed.ncbi.nlm.nih.gov/16940158/ The Nitric Oxide Scavenger Cobinamide Profoundly Improves Survival in a Drosophila Melanogaster Model of Bacterial Sepsis. FASEB J 2006 Sep;20(11):1865-73]

@[https://pubmed.ncbi.nlm.nih.gov/26791145/ Cost of Surviving Sepsis: A Novel Model of Recovery From Sepsis in Drosophila Melanogaster - Intensive Care Med Exp 2016 Dec;4(1):4]

Critique: What is the specificity of DCA? Has it been used in other experiments that investigate the Warburg effect?

Response: DCA has ben used to manipulate metabolic and inflammatory changes in other infection models. Work by Yamane et al did show in a murine model of influenza that both DCA as well as a DCA analog (diisopropylamine dichloroacetate) improved survival (Figure 8) and had similar inhibition of pyruvate dehydrogenase kinase 4 (Table 3). 

#[https://pubmed.ncbi.nlm.nih.gov/24865588/ Diisopropylamine Dichloroacetate, a Novel Pyruvate Dehydrogenase Kinase 4 Inhibitor, as a Potential Therapeutic Agent for Metabolic Disorders and Multiorgan Failure in Severe Influenza. FASEB J 2006 Sep;20(11):1865-73] PLoS One 2014 May 27;9(5):e98032]

Critique: As an output of activation of the immune response (as an approximation of the concept of inflammation? – needs to be explained), only a few AMP genes and Toll are selected. This seems too limited in order to have a good idea about the status of “sepsis survivor” in the treated flies.

Response: As the reviewer rightly suggested, we used the studied AMP’s as an approximation of inflammation based on our earlier research. In our prior work using the same model*, we explored the components of the canonical transcription factor NF-κB (dorsal, Dif, Relish) and a wider panel of receptors, signaling molecules, and antimicrobial peptides (PGRP-SD, Toll, Metchnikowin, Cecropin A, JNK, Drosomycin, Defensin, InR, IRS, PTEN, Akt1, Foxo, mTORC1, and ratio of p-Akt/Akt) as surrogates for sustained inflammation among flies surviving sepsis. 

In our original work*, we observed persistent elevation of NF-κB gene expression as well as activation (Figure 2 in Intensive Care Med Exp 2016) and AMP’s (Figure 3 in Intensive Care Med Exp 2016) in the absence of any obvious infection as shown by culturing flies surviving sepsis.

This was the basis of us establishing the “sepsis survivor” phenotype with the goal of mimicking patients surviving sepsis, yet having ongoing inflammation even at hospital discharge. We distilled our prior experience and used these AMP’s and Toll in the current manuscript. We revised our manuscript accordingly and added the reasoning for the few AMP’s and Toll used.

*[https://pubmed.ncbi.nlm.nih.gov/26791145/ Cost of Surviving Sepsis: A Novel Model of Recovery From Sepsis in Drosophila Melanogaster - Intensive Care Med Exp 2016 Dec;4(1):4]

Critique: The purpose of the geotaxis experiments is not explained and should be elaborated. 

Response: Our goal of using a functional outcome* was to mimic the human experience surviving sepsis. 

*[https://pubmed.ncbi.nlm.nih.gov/26109398/ Physical Activity, Muscle Strength, and Exercise Capacity 3 Months After Severe Sepsis and Septic Shock - Intensive Care Med 2015 Aug;41(8):1433-44]

Critique: The purpose of the experiments of untargeted metabolomics is not explained and should be elaborated. Particular metabolites have the highest weight in principal component analysis: explain and provide evidence. 

Response: Univariate data analysis as shown in Fig. 5 in the revised manuscript and multivariate analysis such as PCA analysis provide different information from the data measured. Univariate analysis suggests that the variable (here a metabolite) presents significantly different amounts in a sample from another sample, control vs. treated for example. Otherwise, multivariate analysis, using all variables in the same calculation gives information about the relationship between variables. 

Here, we have chosen the unsupervised PCA method, which is considered a basic analysis in metabolomics, not susceptible to overfitting, and robust method when clustering samples, as it is the case in Figure 5.

[https://link.springer.com/article/10.1007/s11306-013-0598-6 Reflections on univariate and multivariate analysis of metabolomics data. - Metabolomics. 2014;10(3):361–374.]

 PCA results are evidenced in score-plot figures (Fig. 5) calculated by reducing the number of dimension (or variables) while preserving the initial information (metabolites variability). 

The two principal components of the score plot may be roughly considered as a combination of the initial variables, each of them having a different “weight” in the calculation of the principal component equation. Therefore, the weights of each variable on the PC1 and PC2 describe the metabolites variation responsible for the discrimination of the clusters in the PCA score-plots. In addition, the values of metabolite weight on PC1 and PC2 are given in the table below. 

wpc1 wpc1 Wpc2

Acetate 0.32 0.45

AKG 0.38 0.1

Cit/isoCit -0.35 0.4

Fumarate -0.38 0.08

Lactate 0.3 0.55

Malate -0.35 0.35

Pyruvate 0.38 0.15

Succinate -0.35 0.32

Critique: In the re-infection experiments, unmanipulated flies are compared with sepsis survivors. But sham-treated animals should be used for comparison? 

Response: We agree with the reviewer and eliminated the data on reinfection model from the revised manuscript.

Better description of experiments:

Critique: The number of biological repeats should be clearly mentioned in text and figure legends. The data in the supplementary figures correspond to technical or biological repeats (e.g. qPCR)?

Response: We thank the reviewer, the data in the supplementary figures correspond to biological repeats and added this information into the main text as well as figure legends.

Critique: The criteria for the identification of the targeted metabolites (pyruvate, lactate, TCA intermediates) needs to be described (m/z values, retention times in chromatography…), perhaps as supplementary data.

Response: The methods for LC-MS metabolites vary between each laboratory leading to variations in retention times. In addition, we derivatized the metabolites, so the m/z values will not be the masses of the parent metabolite. For instance, lactate has a mass of 89, but will be 89 + the mass of 3 NP. We did cite the relevant methods in the main text and added a supplemental table (Table S2) as rightly requested.

Critique: Primers and conditions for qPCR need to be provided.

Response: We added a supplementary table with RT-qPCR information.

Critique: Supplementary Figure 6 needs to be explained. There are no legends for the supplementary data in general.

Response: We agree with the reviewer and eliminated the data on reinfection model from the revised manuscript.

Reviewer #2: We thank Prof. Bajgar for the expertise and criticism shared with us

1. Summary of the research and my overall impression

...Even though I like the ideas behind the design of the experiment and building of the story, the data do not entirely adhere to their interpretation and do not support the general conclusions. Therefore, I have to declare here that data seems to me to be over-interpreted ...

Response: We appreciate the candid and constructive input from Prof. Bajgar. We re-worded our discussion remaining loyal to the data present. 

2. Discussion of specific areas for improvement

Major points:

Critique: 1) It is known that bacterial infection causes polarization of macrophages towards aerobic glycolysis in drosophila. Nevertheless, all the detected changes (gene expression and metabolites) have been measured on the whole fly lysates. 

Response: We agree that macrophage polarization is well established in the literature both for mammalian as well as insect models. We think of sepsis as a systemic disease with multiple tissues/cell types involved, so we decided to use whole insect analytes, however we got the inspiration to follow our work with macrophage specific deletion models. 

Critique: Additionally, the treatment affecting the metabolic rearrangement of cells towards oxidative phosphorylation is systemic. The presented data, in my opinion, do not support the metabolic shift towards aerobic glycolysis. The mechanism of the DCA effect on the life span is thus elusive and cannot be interpreted as a Warburg effect reversal. Therefore the mechanism of action of this drug is entirely unknown may influence many different signaling pathways in various tissues and organs. 

Response: We agree with the criticism that one specific pathway for DCA improvement on sepsis. One way to demonstrate this more definitively would have been to do the 13C-glucose labeling experiments that we are discussing now to start after the Covid-19 pandemic normalizes the laboratory environment.

Critique: Moreover, Drosophila - as a model organism, enables to solve particularly the question of tissue-specific effects of such treatment due to versatile genetic tools that were omitted entirely. For me, the whole model of survivors of sepsis and the long-term persisting Warburg effect in flies is a little bit confusing. Authors claim that flies surviving the infection still show increased activation of the immune system and metabolic reprogramming. Nevertheless, in this manuscript, there are completely missing data about the level of bacteria and how long they are persisting in the flies. While trying to find the answer to this question, I found the answer to this question in supplementary data of previously published research publication establishing this experimental model (citation 23). 

Response: Our current data did not show any residual infection after the antibiotic treatment, yet we would also like to remind the reviewers that both the DCA+ as well as the DCA- groups had the same experimental conditions (i.e. infection followed by antibiotics).

Critique: Fig.1 – the number of individuals is not sufficient to claim anything about survivors of infection; it is not clear to me whether more than one biological replicate has been done. 

Response: We had biological replicates and went through this survival analysis with Prof. Chang, one of our co-authors. We agree with the difficulties on performing survival analyses in Drosophila as we keep these animals in vials and one can argue that vials are like clusters. Thus, adding all of the flies together across vials makes it a challenge. For example, two flies in one vial could be regarded to be more alike than one fly from one vial and one from another. We performed Kaplan-Meier survival analyses for the groups adjusting and not adjusting for clusters (vials). The two groups are statistically different (DCA has better survival) at the alpha=0.05 level with and without adjusting. 

The IQR of survival days for the two groups were:

Group A (regular diet): median=12 days (IQR: 7-35 days) and Group B (DCA diet): median=20 days (IQR: 11-58 days). The DCA diet groups showed better survival throughout of the study period, especially after day 12 the separation becomes profound.

Critique: Fig.2 – I would say that there are not enough individuals used in this assay, and the number of individuals in compared groups is strikingly different. 

Response: Our analyses suggested that the lifespan was statistically significant. 

Critique: Fig.3 – Several significant differences do not have displayed the significances in the plots even though they are discussed in the text. Moreover, the data show a striking difference even in sham control groups in comparison to untreated ones. How can the authors explain that the expression level of cecropin-A and drosomycin is increased more than 50-times just in response to sham treatment? 

Response: Figure 3 (now Figure 4) has the data in the figures in the following order: 

Sham regular diet – Sepsis survivors regular diet – Sham DCA diet – Sepsis survivors DCA diet

The 50-fold increase was in response to infection, not to sham treatment.

Critique: Fig.4 – I like the PCA analysis (although it would be nice to see data also in RDA), but have some problems with some interpretations. How is figure 4A, showing no difference between all three groups, support the notion that infection induces the Warburg effect? Why these groups diversify because of DCA feeding (because one would instead expect that DCA treatment would diminish previously observed changes)? 

Response: The Warburg effect is based on the increased of lactate content (which is shown in the Figure 5). The Figure 5A is a multivariate analysis not only based on lactate concentration but on several metabolites aiming to reveal the relationships between the variables measured. The PCA score plot shows that the variability of all metabolites within the 3 groups (unsupervised method) does not reach the clustering of the groups. Nevertheless, when flies are treated with DCA as in figure 5B, the 3 groups tend to separate and when regular diet- and DCA- survivors are compared a clear clustering is obtained in PCA loading plot. 

Critique: Further, the variance is explained mainly by metabolites of the TCA cycle metabolites (as it is mentioned in the results), but the typical changes for the Warburg effect are instead increased glycolysis and pentose phosphate pathway while the TCA cycle undergoes complicated rewiring. 

Response: Most of the metabolites measured as those of the TCA cycle and we agree with the reviewer that TCA cycle undergoes complicated inter-connected modulations as shown in the figures 5 and 6. 

Critique: Are these metabolites changing as well. How are these data in adherence with previously published observation, where for instance, lactate has not been altered in the same treatment? (https://www.mdpi.com/22181989/6/4?view=abstract&listby=date&page_no=1). 

Response: There are a large number of differences between this study and the previous one.

First, the analytical methods are completely different in the physical principles and aims. NMR is a non-targeted method evaluating a large number of variables that will be statistically analyzed (including what could be considered as “noise”). In the present study, the mass spectrometry, far more sensitive, is a targeted method here aimed to measure metabolites of the TCA cycle. 

Secondly, the statistical analyses are completely different as, in the present study, univariate analysis was performed showing the variation of lactate content and unsupervised multiparametric analysis was applied to investigate the possible relationships between these variables. In the previous paper, lactate was not the variable with the highest correlation in the supervised models, as several other metabolites with higher correlation were found. 

As far as the suggested RDA, a supervised method, our number of samples and number of variables could make interpretation difficult (same argument could be made for PLS in this case).

Critique: Fig.5 – According to my opinion, the data are not interpreted correctly since the authors claim that DCA leads to increased TCA cycle rather than the conversion of pyruvate to lactate. Still, there is almost no difference in most of the TCA metabolites in the sham group of flies in response to DCA feeding. 

Response: The figure is now Fig 6. In the sham group, there is no lactate increase and no particular glycolysis over-function, the TCA cycle is running at its usual turn-over. 

Acetyl-CoA did increase with DCA treatment and pyruvate is converted into acetyl-CoA in the mitochondria suggesting that more pyruvate is entering into TCA and less is being converted to lactate. However, as the reviewer rightly suggested, 13C labeling would make this more visual and mechanistic. I think the changes in lactate are not very substantial because there was just so much to start with (mM range). And maybe this is more of a shift from anaerobic to aerobic glycolysis? We overall agree that “Warburg” effect can't be proven with the current studies.

Critique: Fig.6 – The analysis has performed to show whether gain of primed immunity to S. aureus reinfection rely in metabolic rearrangement and can be thus disturbed by DCA feeding. Unfortunately, I have to admit that this experiment failed to show that there is the primed immunity in analyzed flies. The number of individuals in this assay is insufficient. By adding the sham and manipulated group that was excluded from the figure but can be found in data files, it is clear the flies did not gain the primed immunity at all. The rest of the experiment is thus irrelevant. 

Response: We agree with the reviewer, while it is the logical next step to test if trained immunity was activated with the DCA treatment, our current data is not conclusive. We thus eliminated the data and associated interpretations.

Minor points:

Critique: Line4 – I do not entirely understand the last sentence of the abstract, but it may be due to my English.

Response: We modified the last sentence in the abstract to align with the message of the manuscript.

Critique: Introduction – I lack in the introduction the information about what is known about the Warburg effect and polarization of macrophages in Drosophila. Moreover, many of the references are not very well supporting the notions (for example – 17,19, etc.).

Response: Clinicians managing long-term outcomes from sepsis, such as increased risk for mortality, do not have good pre-clinical models to address the clinical questions. Thus, our goal was to have a model relevant to the human experience and we probably used disproportionately more clinical references. However, we agree with Prof. Bajgar using appropriate references in Drosophila and we did add literature on Drosophila macrophage polarization.

Critique: The paragraphs - Fly reinfection and Fly survival and lifespan - should be described more into the detail.

Response: We did eliminate the reinfection data from the revised manuscript. We described the survival and lifespan in more detail.

Critique: Even though there is mentioned the source of primers, it would be valuable to have the sequences of primers and probes used in the appendix. Further, the genes, included in the publication, should be identified by their FLY base FBGN identifier.

Response: We added a supplementary table with qPCR primer information. We did order probes and primers from Applied Biosystems as Assays-on-Demand (AoD). The company provides AoD gene assays with context sequence surrounding the assay location and not the primer sequences.

We added the FBGN identifiers into the manuscript. 

Critique: Discussion – the references should be revisited. The notion on line 331-334 seems to me excessive and directly connected with the published data.

Response: We thank Prof. Bajgar, we revised the references and made the conclusion statement based solely on our data suggesting an association. As both reviewers suggested, this model lends itself for future mechanistic approaches, such as selective genetic manipulation.

---

## [Editor Report · Decision Letter 1]

9 Oct 2020

Dichloroacetate-induced metabolic reprogramming improves lifespan in a Drosophila model of surviving sepsis.

PONE-D-20-05383R1

Dear Dr. Kaynar,

We’re pleased to inform you that your manuscript has been judged scientifically suitable for publication and will be formally accepted for publication once it meets all outstanding technical requirements.

Kind regards,

Fanis Missirlis, Ph.D.

Academic Editor

PLOS ONE
---

## [Editor Report · Acceptance letter]

16 Oct 2020

PONE-D-20-05383R1 

Dichloroacetate-induced metabolic reprogramming improves lifespan in a Drosophila model of surviving sepsis. 

Dear Dr. Kaynar:

I'm pleased to inform you that your manuscript has been deemed suitable for publication in PLOS ONE. Congratulations! Your manuscript is now with our production department. 

Kind regards, 

on behalf of

Dr. Fanis Missirlis 

Academic Editor

PLOS ONE